# A *Bombyx mori* Infection Model for Screening Antibiotics against *Staphylococcus epidermidis*

**DOI:** 10.3390/insects13080748

**Published:** 2022-08-19

**Authors:** Aurora Montali, Francesca Berini, Alessio Saviane, Silvia Cappellozza, Flavia Marinelli, Gianluca Tettamanti

**Affiliations:** 1Department of Biotechnology and Life Sciences, University of Insubria, 21100 Varese, Italy; 2Council for Agricultural Research and Economics, Research Centre for Agriculture and Environment (CREA-AA), 35143 Padova, Italy; 3Interuniversity Center for Studies on Bioinspired Agro-Environmental Technology (BAT Center), University of Napoli Federico II, 80055 Portici, Italy

**Keywords:** *Bombyx mori*, infection model, *Staphylococcus epidermidis*, insect immune response, antimicrobial compounds, glycopeptide antibiotics, vancomycin, teicoplanin, dalbavancin

## Abstract

**Simple Summary:**

The use and misuse of antibiotics in the past decades have contributed to the wide spread of antibiotic resistance, which currently represents a major issue and threat to human health. Consequently, the discovery of new anti-infective molecules is of primary importance. In vivo studies are crucial for testing the efficacy of novel antibiotics. Invertebrate models look promising to reduce the large-scale use of mammalians, which is mainly limited by high costs and ethical concerns. In this scenario, the silkworm proved to be an interesting alternative among insects. Here, we developed a silkworm infection model by challenging the larvae with *Staphylococcus epidermidis*, one common cause of infections in hospitals, and assessing the curative effects of three life-saving glycopeptide antibiotics that are used to treat infections caused by multidrug-resistant Gram-positive pathogens.

**Abstract:**

The increasing number of microorganisms that are resistant to antibiotics is prompting the development of new antimicrobial compounds and strategies to fight bacterial infections. The use of insects to screen and test new drugs is increasingly considered a promising tool to accelerate the discovery phase and limit the use of mammalians. In this study, we used for the first time the silkworm, *Bombyx mori*, as an in vivo infection model to test the efficacy of three glycopeptide antibiotics (GPAs), against the nosocomial pathogen *Staphylococcus epidermidis*. To reproduce the human physiological temperature, the bacterial infection was performed at 37 °C and it was monitored over time by evaluating the survival rate of the larvae, as well the response of immunological markers (i.e., activity of hemocytes, activation of the prophenoloxidase system, and lysozyme activity). All the three GPAs tested (vancomycin, teicoplanin, and dalbavancin) were effective in curing infected larvae, significantly reducing their mortality and blocking the activation of the immune system. These results corroborate the use of this silkworm infection model for the in vivo studies of antimicrobial molecules active against staphylococci.

## 1. Introduction

Antibiotics are considered one of the greatest discoveries of the 20th century and continue to be extremely important in modern medicine, which relies on them both to treat and to prevent infections in high-risk patients [1]. However, the massive use and/or misuse of antibiotics have contributed to the development and spread of resistant bacteria [2]. The need to find and create novel antibacterial compounds is urgent given the increasing problem of antibiotic resistance. In vivo screening of novel antibiotics might accelerate the identification of promising molecules favoring the prioritization of those endowed by an acceptable therapeutic index (i.e., good efficacy versus low toxicity) [3]. The large-scale use of mammalian animal models, primarily mice and rats, at such preclinical phase is not feasible due to the high costs of their handling and maintenance and the ethical considerations that limit their use, as recommended by the European directive on animal protection guided by the 3Rs rules (i.e., Replacement, Reduction, and Refinement) [4]. To develop suitable alternatives, different invertebrate models have been recently proposed, such as amoebas and nematodes [5,6]. Among them, holometabolous insects represent a promising option due to their minimal cost requirements and convenient larval handling [7]. Moreover, the use of insects is not subjected to ethical restrictions [8] and, although they lack acquired immunity, their innate response is evolutionally and functionally analogous to that of mammals [9].

Different insect species, as *Drosophila melanogaster*, *Galleria mellonella*, and *Bombyx mori*, have recently been used to investigate the action of novel antimicrobials, and to test their efficacy [8,10]. A particular advantage in developing silkworm as an infection model is represented by the access to germplasm banks, where *B. mori* is maintained in genetic stock collections; these centers adopt rearing on artificial diet, thus contributing to standardize the quality of this insect supplies [8]. In addition, genome sequencing of *B. mori* [11] has led to the development of molecular tools and gene editing systems, such as RNA interference-based gene silencing, transposon-mediated transgenesis, and CRISPR/Cas9 [12], which proved useful to set up tailored disease models for investigating drug mode of action. Silkworm infection models have been already employed to evaluate antimicrobial drugs against Gram-negative and Gram-positive bacteria of the ESKAPE group (such as *Escherichia coli*, *Pseudomonas aeruginosa*, *Staphylococcus aureus,* and *Klebsiella pneumoniae*) [13,14,15,16], and towards other bacterial pathogens, such as *Francisella tularensis*, *Listeria monocytogenes*, *Bacillus cereus*, and *Mycobacterium abscessus* [17,18,19,20], and fungi (*Candida* spp. and *Aspergillus fumigatus*) [21,22]. However, to date, no study in this setting has been performed by using one of the most common causes of nosocomial infections, the coagulase-negative *Staphylococcus epidermidis*. *S. epidermidis* is an abundant harmless colonizer of human skin and mucosa [23]. However, it becomes an opportunistic pathogen that can cause virulence once it invades the human body via medical and prosthetic devices, such as peripheral or central intravenous catheters or orthopedic implants. Bacteremia arises most commonly by indwelling medical device contamination by *S. epidermidis*, which in addition can produce biofilms resistant to the host defense and antimicrobial treatments [23,24]. 

In the present study, the effects of three life-saving glycopeptide antibiotics (GPAs) (i.e., vancomycin, teicoplanin, and dalbavancin), used in clinical settings to treat severe infections caused by Gram-positive bacteria [25], were investigated in *B. mori* larvae following infection with a clinical isolate of *S. epidermidis*. By monitoring the insect survival rate and the responses of three relevant immunological markers (i.e., hemocyte metabolic rate, lysozyme activity, and phenoloxidase (proPO) system activity), we demonstrated the curative efficacy of the three GPAs, confirming the robustness of this insect-based infection model for testing antimicrobial compounds. 

## 2. Materials and Methods

### 2.1. Experimental Model

The larvae of *B. mori* (polyhybrid (126 × 57) (70 × 90)) used in this study were provided by CREA-AA Sericulture Laboratory (Padova, Italy). Insects were reared in glass Petri dishes at 25 ± 0.5 °C, with 70 ± 5% relative humidity, under a 12L:12D photoperiod. Larvae were fed on artificial diet [26] until the end of the 4th larval instar. After animals had ecdysed to the 5th larval instar, they were synchronized [27] and fed with a germ-free diet [28].

### 2.2. Bacterial Strain

*S. epidermidis* strain 4, a clinical isolate kindly provided by Laboratorio di Microbiologia Clinica-Ospedale di Circolo (Varese, Italy), was grown in 10 mL of Müller–Hinton Broth 2 (MHB2; VWR International Srl, Radnor, PA, USA), overnight at 37 °C under 200 rpm shaking. Cells harvested by centrifuging 1 mL of culture at 1900× *g* at 4 °C for 10 min were then resuspended at the desired concentration with saline solution (0.6% *w*/*v* NaCl). To determine the volume of saline solution to be used, the optical density of the culture was measured, considering that one unit of OD_600nm_ corresponded to 7.03 × 10^8^ CFU (colony forming units)/mL.

### 2.3. Minimum Inhibitory Concentrations (MICs) and Minimum Bactericidal Concentrations (MBCs)

MICs of vancomycin (Sigma-Aldrich, St. Louis, MO, USA), teicoplanin (Sigma-Aldrich), and dalbavancin (kindly provided by Sanofi, Brindisi, Italy) towards *S. epidermidis* strain 4 were determined by the broth dilution method, following Clinical and Laboratory Standards Institute guidelines [29]. A total of 5 × 10^5^ viable bacterial cells were inoculated into MHB2 medium (VWR International Srl) in 96-well plates and supplemented with increasing antibiotic concentrations (from 0 to 128 µg/mL) up to 100 µL final volume. Plates were then incubated for 16–20 h at 37 °C and 100 rpm. MIC was defined as the minimal concentration of antibiotic at which no turbidity could be detected. For MBC determination, bacterial cultures used for the MIC test were plated onto Müller–Hinton Agar (MHA; VWR International Srl) and incubated at 37 °C for 24 h. MBC was defined as the lowest concentration of antibiotic at which no visible growth was detected on plates. All experiments were performed at least in triplicate.

### 2.4. Injection of Larvae and Collection of Hemolymph

Insects were infected at the second day of the fifth larval instar by injecting bacteria in the second right proleg with autoclaved Hamilton 1702 LT 25 µL syringes (Hamilton Company, Reno, NV, USA), under a sterile hood. After the injection, larvae were reared at 37 °C. Hemolymph collection was performed by puncturing the larvae or cutting the second left proleg 6 or 24 h after the infection depending on the analysis.

### 2.5. Lethal Dose 50 (LD_50_) for S. epidermidis 

To determine the LD_50_, larvae were infected with *S. epidermidis* at different concentrations (from 3 × 10 CFU to 3 × 10^5^ CFU in a final volume of 10 µL). The mortality rate was monitored every 24 h for three days. Larvae were considered dead when no reaction to a stimulation with a plastic tip was observed. Uninjected larvae and larvae injected with 10 μl of saline solution (0.6% *w*/*v* NaCl) were used as controls. Forty larvae for each experimental condition were used. LD_50_, calculated by Probit analysis [30], was defined as the concentration of bacteria at which 50% of animals died within 3 days after the infection. 

### 2.6. Administration of Antibiotics

The effects of GPA administration were evaluated by injecting 10 µL of *S. epidermidis* at LD_50_ into the larvae, and 10 µL of vancomycin, teicoplanin, or dalbavancin (at a concentration equal to 8.75 µg/g body weight) after 2 h, as previously reported [15]. Control groups were represented by uninjected larvae, larvae injected once or twice with saline solution (10 µL of 0.6% *w*/*v* NaCl eventually repeated after two hours), and healthy larvae injected with 10 µL of antibiotic. Larval mortality was monitored for three days. Fifty larvae were used for each experimental condition.

### 2.7. Analysis of the Immunological Markers

Untreated larvae, larvae injected once or twice with 10 µL of 0.6% *w*/*v* NaCl, larvae injected with 10 µL of antibiotic (i.e., vancomycin, teicoplanin, or dalbavancin at 8.75 µg/g body weight), larvae injected with 3 × 10^3^ CFU of *S. epidermidis* in 10 µL, and larvae injected with 3 × 10^3^ CFU of *S. epidermidis* in 10 µL and two hours later with 10 µL of antibiotic (8.75 µg/g body weight) were used for the evaluation of the immunological markers (i.e., cells viability, lysozyme, and prophenoloxidase activity).

#### 2.7.1. Hemocyte Viability

Hemolymph was collected from larvae 24 h after the first injection and diluted 1:50 with Saline Solution for Lepidoptera (sucrose 210 mM, KCl 45 mM, Tris-HCl 10 mM, pH 7.0). CellTiter-Glo Luminescent Cell Viability Assay (Promega, Madison, WI, USA) was used to analyze the hemocyte viability. A total of 100 µL of CellTiter-Glo were added to 100 μl of diluted hemolymph into a 96-well plate, and then incubated for 5 min at room temperature on an orbital shaker. The measurement of luminescence was performed by using an Infinite F200 96-well plate-reader (Tecan, Männedorf, Switzerland). Ten surviving larvae for each experimental group were analyzed.

#### 2.7.2. Prophenoloxidase (proPO) System Activation

Hemolymph was collected from pools of three larvae and centrifuged at 250× *g* for 5 min at 4 °C. A total of 100 µL of hemolymph were loaded into 96-well plates and the absorbance was measured by reading OD_450nm_ every 10 min for 50 min using an Infinite F200 96-well plate-reader (Tecan) [31]. A total of 100 µL of hemolymph supplemented with 2.5 mM N-phenylthiourea (PTU, Sigma-Aldrich) dissolved in 100% *v*/*v* EtOH were used as negative controls. The activation of proPO system was evaluated by incubating hemolymph with β-glucans from *Saccharomyces cerevisiae* (Zymosan, Sigma-Aldrich) and 4 mM CaCl_2_ [32], too. ΔOD_450nm_ was calculated subtracting the OD_450nm_ recorded at time zero from the values obtained at the different time points. To evaluate the melanization rate, linear regression was performed for each ΔOD_450nm_ measurement obtained versus time [31]. 

#### 2.7.3. Lysozyme Activity

Hemolymph was extracted from a pool of three larvae 6 h after infection and added with a few crystals of PTU to avoid melanization. After two centrifugations at 250× *g* for 5 min and one at 1600× *g* for 10 min at 4 °C, the supernatant was collected and diluted 1:10 with sterile phosphate-buffered saline (PBS: 1370 mM NaCl, 27 mM KCl, 100 mM Na_2_HPO_4_ × 12H_2_O, 19.8 mM KH_2_PO_4_). The activity of lysozyme was determined according to Bruno et al. [33]. Briefly, 100 µL of diluted hemolymph were added to 150 µL of 0.45 mg/mL lyophilized *Micrococcus lysodeikticus* (Sigma-Aldrich) in 30 mM phosphate buffer (38 mM KH_2_PO_4_, 61.4 mM K_2_HPO_4_, pH 7.2) (OD_600nm_ of 0.6–0.7). *M. lysodeikticus* and hemolymph both added with PBS were used as controls. Finally, absorbance at 450 nm was recorded every 30 s for 10 min by using an Infinite F200 96-well plate-reader (Tecan, Switzerland).

### 2.8. Statistical Analysis

Statistical analysis was performed using ANOVA, followed by Tukeyʹs Honestly Significant Difference (HSD) test (significance *p* < 0.05).

## 3. Results

### 3.1. Larval Survival and Calculation of LD_50_

To determine the LD_50_ for infection experiments, *B. mori* larvae were injected with increasing concentrations of *S. epidermidis*. Daily monitoring revealed a normal development of control groups (i.e., untreated larvae and larvae injected with 0.6% *w*/*v* NaCl), with a survival rate of 100% after 72 h (Figure 1). A correlation between the mortality rate of larvae and the concentration of injected bacteria was instead observed after *S. epidermidis* injection. In detail, 70% of the larvae survived 72 h after the injection of 3 × 10^2^ CFU of *S. epidermidis* (Figure 1), while only 27% and 11% of insects were still alive after the infection with 3 × 10^3^ and 3 × 10^4^ CFU of bacteria, respectively. Finally, no larvae infected with 3 × 10^5^ CFU survived after 72 h (Figure 1). The lethal bacterial dose that killed 50% of the infected larvae (LD_50_), calculated with Probit analysis, proved to be 1.05 × 10^3^ CFU.

### 3.2. Effect of GPA Administration to the Larvae 

The curative efficacy of the three selected GPAs (i.e., vancomycin, teicoplanin, and dalbavancin) was assessed by administering the antibiotics to the larvae infected with *S. epidermidis* at LD_50_. Larvae injected with only GPAs were monitored to exclude any toxic effect. Moreover, insects were injected twice with saline solution to exclude any potential side effect due to the double injection (bacteria plus antibiotic). In accordance with previous observations [15], all the control groups showed 100% survival (Figure 2) confirming that GPAs were not toxic to larvae and that the experimental procedure of injection was reliable. All the GPAs tested improved the survival rate of infected silkworms. Indeed, while only 51% of the infected larvae survived 72 h after the infection, the administration of vancomycin, teicoplanin, and dalbavancin increased the survival rate to 90%, 93%, and 97%, respectively (Figure 2).

The minimum inhibitory concentration (MIC) and the minimum bactericidal concentration (MBC) of the three GPAs towards *S. epidermidis* (Table 1) were in agreement with the results reported above. In fact, the selected strain was sensitive to all three GPAs and dalbavancin proved to be, among the three of them, the most effective antibiotic also in vitro. 

### 3.3. Analysis of the Immunological Markers

The insect immune system consists of a fine and balanced crosstalk between hemocytes and humoral molecules, which aims at maintaining the hemolymph devoid of pathogens. Herein, we evaluated the cellular and humoral responses of *B. mori* larvae following *S. epidermidis* infection.

#### 3.3.1. Hemocyte Viability

Hemocytes are the main mediators of the cellular immune response in insects [34]. To evaluate the recruitment/activation of these immune cells following pathogen invasion, their viability was evaluated by quantifying the ATP content through a luminescence assay. In insects infected by *S. epidermidis,* a significant increase in luminescence was observed, whereas the treatment with the three GPAs restored the ATP amount to levels comparable to those of the control groups (Figure 3). These results indicated that the antibiotics blocked the infection and, consequently, the involvement of hemocytes in the immune response. 

#### 3.3.2. Activation of Prophenoloxidase System 

In insects, the entrance of pathogens in the hemocoel can induce the activation of the proPO cascade, resulting in the production of melanin to isolate the foreign agents [35]. No significant variation in the activity of proPO system was instead observed after the infection of larvae with 3 × 10^5^ CFU of *S. epidermidis*, compared to the control (Figure 4). Given these results, Zymosan, a specific activator of the proPO system, was used in combination with CaCl_2_ to check if the enzyme activity responded to specific stimuli. A marked increase in the absorbance values confirmed a consistent activation of phenoloxidase under these last conditions (Figure 4), contrarily to what observed following bacterial infection.

#### 3.3.3. Lysozyme Activity

Lysozyme cleaves the peptidoglycan component of the bacterial cell wall, leading to microbial cell death. The production of this enzyme generally increases after the pathogen attack [36]. In uninjected larvae, as well as in other control groups (i.e., healthy larvae injected once or twice with 0.6% *w*/*v* NaCl or with the antibiotics), the activity of lysozyme was comparable. The enzyme activity markedly increased after bacterial infection (Figure 5). Lysozyme activity was restored to the basal levels of the control groups following the administration of vancomycin, teicoplanin, or dalbavancin (Figure 5). 

## 4. Discussion

Various invertebrate models have been proposed over the years to overcome the drawbacks associated with the use of mammals in drug discovery and development, which include not only the ethical and regulatory issues, but also the high costs for animal handling, and the need of ad hoc facilities for their maintenance [37,38]. In this scenario, the silkworm *B. mori* has emerged as a suitable model for mimicking infections by various human pathogens, including bacteria [13,15,39,40,41] as well as filamentous fungi [22] and yeasts [39]. Moreover, this insect has been recently used to test the efficacy of antibacterial agents, such as kanamycin, tetracycline, fluconazole, and vancomycin, in curing infections [7,13,15,39,42].

The optimal temperature for rearing silkworms is commonly 25 °C [26], but this condition significantly differs from the human body temperature. The resistance of *B. mori* larvae to higher temperatures (37 °C) for a limited period of time without any apparent side effect was recently proved [15]. This feature supported the development of a silkworm infection model, which can operate at the human body temperature, and better mimics in this way the trend of bacterial infections in humans [7,15,42]. Although a few papers reported on *B. mori* infected with the coagulase-positive *S. aureus* [13,15,39,40,41], to our knowledge, no publication has previously described infections determined by coagulase-negative staphylococci (CoNS), such as *S. epidermidis* and *S. haemolyticus*. Opportunistic CoNS, and *S. epidermidis* above all, are emerging as major sources of nosocomial infections, especially of foreign body-related infections (FBBIs) associated with indwelling or implanted devices and transmitted by medical and/or nursing procedures [43,44]. *S. epidermidis*-related infections include prosthetic valve endocarditis, keratitis associated with contact lens use, and intravascular catheter- and prosthesis-associated infections. Therapeutic options against CoNS are limited, due to the large number of methicillin-resistant strains or strains with reduced susceptibility to quinolones or to GPAs. In addition, these staphylococci form multilayered and difficult-to-eradicate biofilms, which are highly impenetrable to the majority of antibiotics [43,44].

In our work, we reported on the silkworm model infected at 37 °C with a clinical isolate of *S. epidermidis* and, to validate it, we investigated the curative effect of three molecules belonging to the GPA class. GPAs are defined as ‘drugs of last resort’ for their role in treating life-threating infections caused by staphylococci, enterococci, and *Clostridium difficile* [25]. Their use in fighting clinical infections by *S. epidermidis* is consolidated, too [45]. The selected molecules for this study included the first-generation GPAs vancomycin and teicoplanin, introduced in 1958 and 1988, respectively, and still largely used in clinics, and the second-generation dalbavancin, approved in 2014 and designated as Qualified Infection Disease Product by the Food and Drug Administration (FDA) for its potency and extended dosing interval [25]. We determined the susceptibility profile of our *S. epidermidis* clinical isolate to the three selected GPAs by measuring their MICs and MBCs following in vitro standard protocols. Dalbavancin showed the highest efficacy in inhibiting the growth of the target strain, with MIC and MBC values lower than those recorded for vancomycin and teicoplanin, as also demonstrated in previous publications [46,47,48,49]. Consistently with previous data [46], teicoplanin was confirmed to have a weaker bactericidal effect on *S. epidermidis* cells than the other two GPAs. As regards the in vivo infection model, the dosage of antibiotics administered to the larvae (i.e., 8.75 µg/g body weight) was determined according to Montali et al. [15] and was comparable to the amount of vancomycin used to treat severe staphylococcal infections in humans. It is noteworthy that this dosage, as well as higher concentrations of all three GPAs (up to 35 µg/g body weight), did not exert any significant toxicity in the larvae [15]. Under the tested conditions, all the three antibiotics proved to be effective in counteracting the infection caused by our *S. epidermidis* clinical isolate.

Previous studies explored the use of the silkworm as infection model mainly monitoring the survival rate of the larvae and, only in some cases, evaluating additional markers, such as the quantification of bacteria and/or their localization in the insect body [13,39,42]. However, detailed studies on the immune system of *B. mori* [50] might provide new tools to finely dissect the physiological mechanisms triggered by invading bacteria. Although additional markers (e.g., analysis of the bacterial load and expression of antimicrobial peptides) were monitored in our previous study [15], the complexity of the procedure and the limited reproducibility of the results obtained by following their temporal trend, prompted us to look for new ones. Since our final goal was to use the silkworm model in the early antibiotic discovery and development phase as a means to select molecules with a promising therapeutic index, we looked for robust and trustable markers that could be easy and rapid to monitor. Our results demonstrate that the administration of GPAs in infected silkworm stopped the activation of the insect’s immune response and that the protective effect of these antibiotics could be appreciated at different levels. First, hemocytes, which play a primary role against the invasion of microorganisms in insects [51], confirmed to be a reliable marker, showing a reduced activity in larvae treated with GPAs, caused by a block in their recruitment/activation [15]. In relation to the humoral response that cooperates with the cellular processes to maintain the hemolymph devoid of pathogens [52], we examined the possible induction of two different enzyme activities, i.e., lysozyme and proPO. Lysozyme hydrolyzes peptidoglycan β-(1,4) glycosidic bonds in the cell wall of Gram-positive bacteria [52]. Our results confirm the activation of this enzyme following the injection of *S. epidermidis* in the larvae [53] and its return to control levels after GPA administration, demonstrating the antibiotic efficacy in reducing the effects of bacterial infection. To the best of our knowledge, this marker was used herein for the first time in an insect infection model. It is a rapid, simple, and reliable assay, which could be thus recommended for further studies on antibiotic response in insects. On the contrary, no difference in proPO activity between the control and infected larvae was observed, suggesting that this marker is not suitable for such type of investigations. In insects, proPO leads to melanin formation by generating quinones and other reactive intermediates, with the final aim of trophically isolating and killing pathogens and parasites [54]. The lack of activation of this enzymatic cascade might depend on the bacterial load in the hemolymph, since it is known that proPO system is induced only over a certain threshold concentration of bacteria [33] to avoid the unneeded production of melanin, which is toxic for the insect itself [55]. An alternative possible explanation might involve the ability of some specific pathogens to produce serpin-type inhibitors or non-proteinaceous factors (e.g., polyphenol derivatives), which specifically interfere with the proteolytic activation of proPO, increasing their survival in the hosts [54,56]. Further investigations in this direction would be useful to understand the role of proPO system following bacterial infections.

In conclusion, this study paves the way to use silkworm as a trustable infection model for testing antimicrobial compounds with therapeutic potential against staphylococci. In addition to monitoring the larval survival rate, two additional immunological markers were validated (i.e., hemocyte viability and lysozyme activation) and a third one proved useless (proPO system) for this purpose. The use of these markers could allow to better analyze positive or negative responses of the silkworm to the administration of different antimicrobial products, although a preliminary and accurate verification of their behavior at different temperatures is recommended before introducing them into the screening protocol. Moreover, the reduced number of used insects and the reduced time required for the analysis of these markers could lead to identify new tools useful to improve the procedures for screening antimicrobial compounds on large scale. Our final hope is that this insect infection model might be helpful to accelerate the discovery and development of novel compounds that are urgently needed to contrast the antibiotic resistance.

## Figures and Tables

**Figure 1 insects-13-00748-f001:**
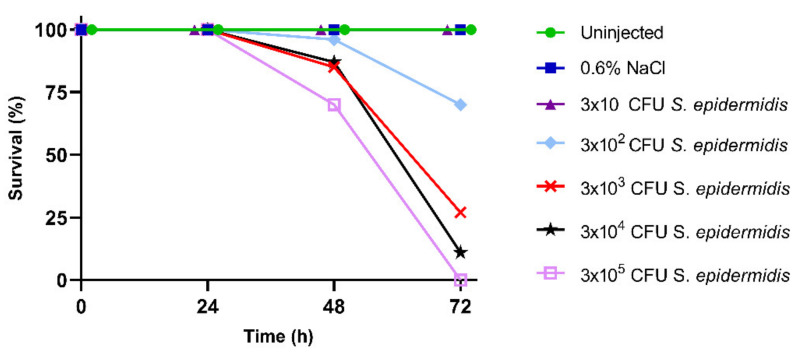
Survival rate of *B. mori* larvae injected with increasing CFU (colony forming unit) of *S. epidermidis*.

**Figure 2 insects-13-00748-f002:**
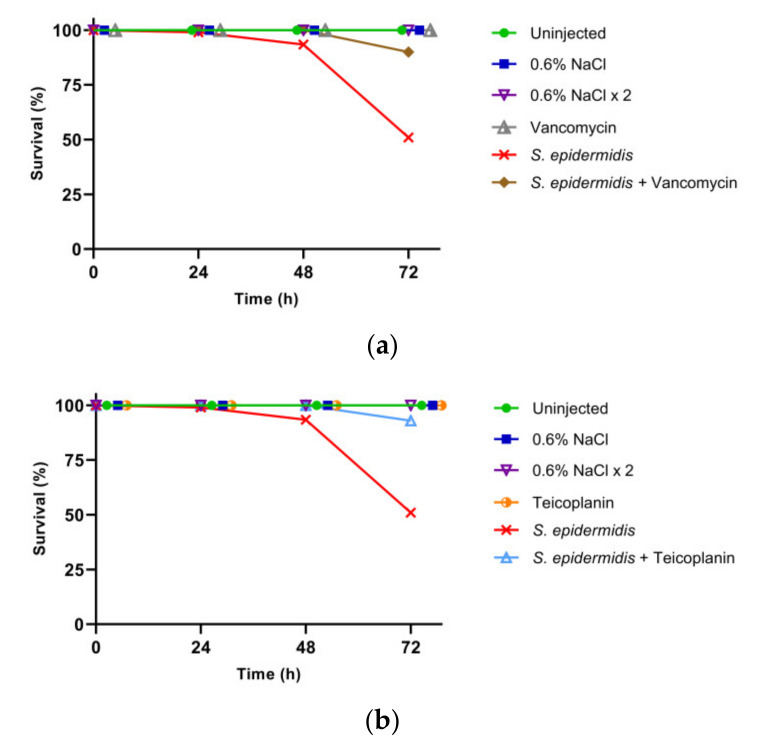
Survival rate. Curative efficacy of vancomycin (**a**), teicoplanin (**b**), and dalbavancin (**c**) (8.75 µg/g larval body weight) on larvae infected with *S. epidermidis* (1.05 × 10^3^ CFU). Results are from the same experiment run in parallel, using the same control groups (uninjected larvae and healthy larvae injected once or twice with saline solution) and infected larvae.

**Figure 3 insects-13-00748-f003:**
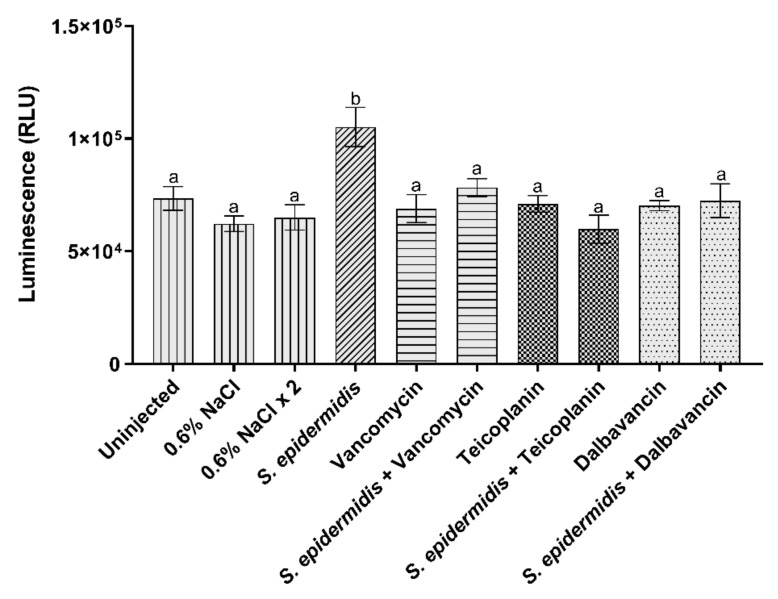
Hemocytes activity detected using a luminescence assay. Values represent mean ± s.e.m. Different letters indicate statistically significant differences among treatments (*p* < 0.05).

**Figure 4 insects-13-00748-f004:**
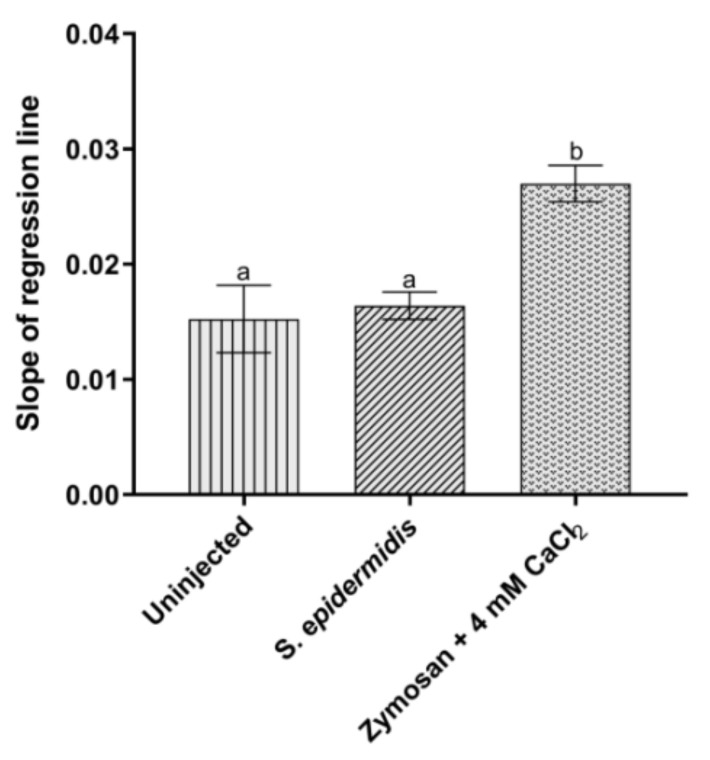
Analysis of proPO system activation. Values represent mean ± s.e.m. Different letters indicate statistically significant differences among treatments (*p* < 0.05).

**Figure 5 insects-13-00748-f005:**
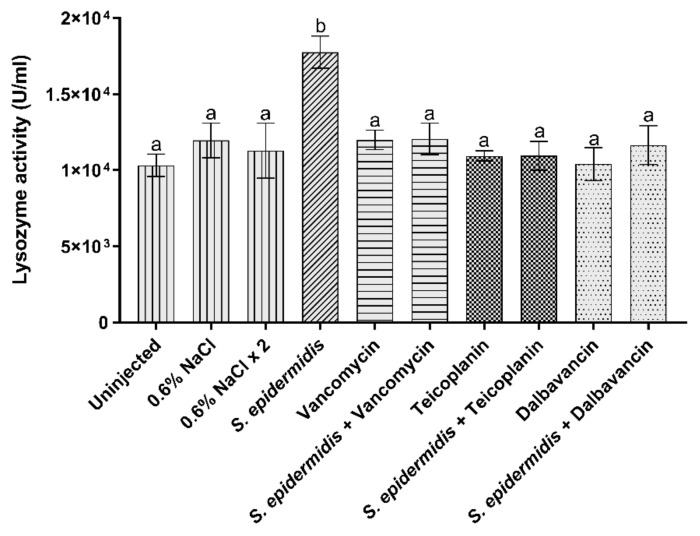
Lysozyme relative activity. Values represent mean ± s.e.m. Different letters indicate statistically significant differences among treatments (*p* < 0.05).

**Table 1 insects-13-00748-t001:** Minimum inhibitory concentration (MIC) and minimum bactericidal concentration (MBC) of vancomycin, teicoplanin, and dalbavancin towards *S. epidermidis*. Values represent the average of data from three independent experiments.

Antibiotic	MIC (µg/mL)	MBC (µg/mL)
Vancomycin	2	16
Teicoplanin	1	32
Dalbavancin	0.25	8

## Data Availability

The datasets generated for this study are available on reasonable request to the corresponding author.

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
