# Peer review of "A Bombyx mori Infection Model for Screening Antibiotics against Staphylococcus epidermidis"

_insects, 2022, doi:10.3390/insects13080748_

Round 1

Reviewer 1 Report

The manuscript entitled “A Bombyx mori infection model for screening antibiotics against Staphylococcus epidermidis” examines to establish a silkworm infection model with Staphylococcus epidermidis for evaluating the efficacy of antibiotics in vivo. The administration of S. epidermidis (3 x 105 cells) led to silkworm death within 72 h after injection. The LD50 values were determined using the silkworm infection model. Administration of antibiotics such as vancomycin, teicoplanin, and dalbavancin was effective to inhibit the silkworm death caused by the injection of S. epidermidis. Lysozyme activity in silkworm hemolymph was increased by the injection of S. epidermidis. The lysozyme activity induced by S. epidermidis was inhibited by the administration of antibiotics. These results suggest that the silkworm infection model with S. epidermidis was established for the evaluation of antibiotics.

The study includes beneficial information to identify the novel antibiotics using the in vivo evaluation system. This is the first report to establish the silkworm infection model with low biosafety level Staphylococci. Therefore, the infection model is useful to evaluate antibiotic candidates without high-level biosafety settings.

The manuscript is well written.

Thank you very much for giving me a chance to review the manuscript.

Author Response

Response to Reviewer 1 Comments

The manuscript entitled “A Bombyx mori infection model for screening antibiotics against Staphylococcus epidermidis” examines to establish a silkworm infection model with Staphylococcus epidermidis for evaluating the efficacy of antibiotics in vivo. The administration of S. epidermidis (3 x 10 cells) led to silkworm death within 72 h after injection. The LD values were determined using the silkworm infection model. Administration of antibiotics such as vancomycin, teicoplanin, and dalbavancin was effective to inhibit the silkworm death caused by the injection of S. epidermidis. Lysozyme activity in silkworm hemolymph was increased by the injection of S. epidermidis. The lysozyme activity induced by S. epidermidis was inhibited by the administration of antibiotics. These results suggest that the silkworm infection model with S. epidermidis was established for the evaluation of antibiotics. The study includes beneficial information to identify the novel antibiotics using the in vivo evaluation system. This is the first report to establish the silkworm infection model with low biosafety level Staphylococci. Therefore, the infection model is useful to evaluate antibiotic candidates without high-level biosafety settings. The manuscript is well written. Thank you very much for giving me a chance to review the manuscript.

We thank the Reviewer for appreciating our study.

Reviewer 2 Report

The manuscript described a novel silkworm model infected with S. epidermidis and evaluated the glycopeptide antibiotics efficacy by several indicators. The results were highly interesting and valuable findings for antimicrobial development and pathogenicity analysis of pathogens.

The authors evaluated the pathogenicity and immunological response of silkworm against the infection at 37 ËšC, however, usually 25-30ËšC is a physiological condition for silkworm rearing, as the authors pointed out. It is required to test the pathogenicity of S. epidermidis and the immunological response of silkworm. Because silkworm immune response and pathogenicity of the bacteria often change due to temperature differences.

In addition, the bacteria dynamics in the silkworm have to be analyzed. The time course of the number of bacteria in the hemolymph is needed to understand how the bacteria kills silkworms and antibiotics treat the silkworm infected with the bacteria. Furthermore, this information is helpful in understanding why the immune responses, such as hemocyte viability, phenol oxidase, and lysozyme activity, were different among the groups.

Further, the authors discussed the therapeutic efficacy of glycopeptide antibiotics in silkworms infected with S. epidermidis and compared MIC among these antibiotics. However, there is no link between them. To link these results, it is better to evaluate effective doses 50 (ED50) and discuss the differences between MIC and ED50 relationship. In addition, the authors claimed that the other marker is useful for evaluating the efficacy of antibiotics. Although, the easiest way to assess the effectiveness of antibiotic treatment is to monitor survival rates simply. Thus, the authors need to present more merit to analyze these immune responses after antibiotic treatment.

Author Response

Response to Reviewer 2 Comments

The manuscript described a novel silkworm model infected with S. epidermidis and evaluated the glycopeptide antibiotics efficacy by several indicators. The results were highly interesting and valuable findings for antimicrobial development and pathogenicity analysis of pathogens.

Thank you very much for appreciating our study.

Point 1: The authors evaluated the pathogenicity and immunological response of silkworm against the infection at 37°C, however, usually 25-30°C is a physiological condition for silkworm rearing, as the authors pointed out. It is required to test the pathogenicity of S. epidermidis and the immunological response of silkworm. Because silkworm immune response and pathogenicity of the bacteria often change due to temperature differences.

We are aware that 25 °C is the optimal temperature for rearing silkworms, and that such condition has been used in the past to evaluate the immune response of Bombyx mori to different bacterial infections (Kaito et al., 2002; Hamamoto et al., 2004; Kaito et al., 2005; Barman et al., 2008; Uchida et al., 2014). However, as reported in our previous article (Montali et al., 2020) and remarked in this manuscript, this rearing temperature is significantly distant from the human body temperature which is around 37 °C. Moreover, staphylococcal proliferation, gene expression, and protein production (including virulence-related factors) are known to be influenced by incubation temperature (Bastock et al., 2021). For these reasons, since the present study did not aim to evaluate in detail the physiological response of the silkworm to infections at different temperatures, but indeed to develop a robust insect model able to properly mimic bacterial infections in humans, we decided to operate at 37°C. We better clarified this aspect in the text (lines 281-287). 

Point 2: In addition, the bacteria dynamics in the silkworm have to be analyzed. The time course of the number of bacteria in the hemolymph is needed to understand how the bacteria kills silkworms and antibiotics treat the silkworm infected with the bacteria. Furthermore, this information is helpful in understanding why the immune responses, such as hemocyte viability, phenol oxidase, and lysozyme activity, were different among the groups.

The development of the infection model proposed in this study is based on the use of easy-to-perform and relatively cheap markers. As reported in our previous article (Montali et al., 2020), the determination of the time course of the bacterial load in the hemolymph requires a lot of time, efforts, and materials. Moreover, in that study we demonstrated that the quantification of the number of bacteria in the hemocoel requires numerous replicas to obtain robust and reliable results, thus this marker does not appear adequate for screening and evaluating new candidate antimicrobial compounds, especially on a large scale. For the same reason, we did not consider here qPCR analysis of AMP expression, a marker that we previously tested (Montali et al., 2020). Taking this information into account, in the present manuscript our efforts were focused on easy-to-analyze markers (i.e., survival rate, hemocyte viability, proPO activation) and on testing new ones for the first time, such as lysozyme activity, to improve the infection model that we established in a previous study (Montali et al., 2020). Following the reviewer comment, we better clarified our approach in the text (lines 327-333).

Point 3: Further, the authors discussed the therapeutic efficacy of glycopeptide antibiotics in silkworms infected with S. epidermidis and compared MIC among these antibiotics. However, there is no link between them. To link these results, it is better to evaluate effective doses 50 (ED50) and discuss the differences between MIC and ED50 relationship.

We thank the Referee for the suggestion. However, as highlighted also above, the main objective of this study was to validate a simple but reliable silkworm infection model to be used for a preliminary screening in vivo of molecules with a proved antimicrobial activity in vitro and, in particular, to establish if: i) they are potentially toxic for animals and ii) they can cure S. epidermidis infection in vivo. For these reasons we did not focus the attention on the mode of action of the antibiotics, by considering the median effective dose (ED50), nor we investigated pharmacokinetics and pharmacodynamics, or GPA administration procedures. Indeed, the three GPAs considered in this study present different protocols of administration in humans while, in the infection model that we are developing, this procedure is simplified and limited to a single injection of GPA at a defined dose (as reported in the manuscript). In our case, the determination of MIC was used to validate the sensitivity of the bacterial strain employed in this study to the GPAs, by using in vitro standard methods. The results were used as a starting point to design the subsequent analyses. This aspect is now better clarified in the text (lines 308-310).

Point 4: In addition, the authors claimed that the other marker is useful for evaluating the efficacy of antibiotics. Although, the easiest way to assess the effectiveness of antibiotic treatment is to monitor survival rates simply. Thus, the authors need to present more merit to analyze these immune responses after antibiotic treatment.

We agree with the referee that the most direct evidence to assess the antibiotic effectiveness is the analysis of the survival rate of the larvae, but the use of immunological markers could help to identify processes that are related to a positive or negative response by the silkworm to the administration of antimicrobial compounds, thus opening new perspectives to the use of infection models based on insects. This is one of our goals for the next future that, as reported above, will require an in-depth analysis of the immune response of the silkworm at higher temperatures. Another important aspect worthy to be underlined from an applied perspective is that the reduced number of larvae used for the immunological markers, as well as the reduced time for their analysis (e.g., 6 hours of incubation with bacteria for lysozyme activity), and high reproducibility of results compared to the evaluation of the survival rate, could offer the possibility to identify one or more additional markers to improve the procedures for screening antimicrobial compounds on large scale with this insect-based system. This aspect was reinforced in the revised text (lines 366-372). 

Round 2

Reviewer 2 Report

I understand the author's rebuttal regarding the author's intent on this manuscript. The manuscript was revised adequately and is suitable for publication.